# Light-Activated Room Temperature Surface Acoustic Wave H_2_S Sensor Based on Bi_2_S_3_ Nanoribbons

**DOI:** 10.3390/s25041122

**Published:** 2025-02-12

**Authors:** Chong Li, Sami Ramadan, Hao Kan, Lina Wang

**Affiliations:** 1School of Electronic Engineering, Huainan Normal University, Huainan 232038, China; chongli@hnnu.edu.cn; 2Department of Materials, Imperial College London, London SW7 2AZ, UK; 3School of Information Science and Engineering, University of Jinan, Jinan 250022, China

**Keywords:** surface acoustic wave (SAW), Bi_2_S_3_, nanoribbons, H_2_S, light-activated sensor

## Abstract

The expansion of the Internet of Things (IoT) has rendered wireless passive, highly stable, and room-temperature gas sensors indispensable for sensor applications. In this work, a room-temperature surface acoustic wave (SAW) H_2_S sensor based on a thin film of nano-mesh woven with Bi_2_S_3_ nanoribbons was successfully designed and prepared. The impact of varying inorganic salts solution ligand substitution of long-chain organic ligands of Bi_2_S_3_ films on performance was assessed. Notably, the responses of the sensors following ligand substitution exhibited improvement to varying degrees. In particular, the Cu(NO_3_)_2_-treated sensor to 10 ppm H_2_S was 203% of that of the untreated sensor. Furthermore, the impact of visible light activation on sensor performance was assessed. The results show the sensor has a high sensitivity to H_2_S molecules under yellow light activation at room temperature, with excellent selectivity, fast response speed and low detection limit. The sensor exhibited a response to 10 ppm H_2_S under yellow light activation that was approximately equal ~ two times greater than the response observed in a dark environment. This work provides a novel approach to enhance the performance of room-temperature SAW H_2_S sensors.

## 1. Introduction

Hydrogen sulfide (H_2_S) is a toxic, corrosive and irritating odor gas that is commonly found in human daily life and industrial production [1]. High concentrations of H_2_S have been reported to cause damage to the animal/human nervous system, thus posing a threat to animal/human health [2]. In addition, H_2_S is a potential marker for certain diseases in human exhaled breath [3]. The monitoring of trace amounts of H_2_S in exhaled gas has been shown to provide useful diagnostic information for potential diseases (such as respiratory diseases, metabolic disorders) [4]. Consequently, there is an increasing demand for the detection of low H_2_S concentrations at room temperature in many fields, including chemical, industrial and medical diagnostics.

Advances in technology have resulted in the development of a wide variety of gas sensors, including semiconductor [5], electrochemical [6], infrared optical [7], quartz crystal microbalance (QCM) [8] and surface acoustic wave (SAW)-type sensors [9]. Among them, surface acoustic wave (SAW) sensors have been extensively utilized for the detection of gases, humidity and biological substances due to their high stability and easy integration [10,11]. Furthermore, SAW devices have the capacity to be wireless and passive, with the possibility of supporting remote sensing, as well as stable operation under harsh conditions [12]. Consequently, SAW gas sensors are expected to provide fast and selective detection of H_2_S sensors operating wirelessly at room temperature for the next generation of the Internet of Things (IoT).

For gas sensors, the selection of a suitable material is paramount to the effectiveness of the sensor [13]. The material must exhibit high selectivity and high adsorption capacity for the target gas to ensure an accurate and reliable sensor. Recent years have witnessed a rapid development in the use of various low-dimensional materials deposited on the surface of SAW devices for the purpose of gas detection, such as polymers [14], metal oxide nanomaterials [15] and various carbon-based nanomaterials [10,16]. However, the majority of these materials are incompatible with high sensing response and fast response at room temperature, thus limiting their practical application in real-time gas sensing.

The utilization of light activation energy to enhance the gas adsorption/desorption kinetics of SAW sensors can result in sensors with high sensitivity and expedited response/recovery speed [17]. For instance, Jakubik et al. reported a light-activated SAW sensor based on photoconductive polymer films for DMMP sensing prepared by spray deposition of polymers on the surface of SAW devices [18]. Pasupuleti et al. reported an SAW for NH_3_ gas detection under UV irradiation at room temperature [19]. Under UV irradiation, the ZnO@MXene SAW sensor shows significant improvement in frequency response and selectivity. Zhang et al. reported a GO-MoS_2_/SnO_2_ composite film for the detection of NO_2_ at room temperature under ultraviolet (UV) light irradiation [20]. The key sensing mechanism is mainly the generation of a large number of photoelectron–hole pairs under UV light irradiation, which accelerates the adsorption and reaction process of NO_2_ molecules.

However, the majority of reports on light-activated SAW gas sensors have focused on NO_2_ and NH_3_ gases. There are no other reports on light-activated SAW H_2_S sensors operating at RT using visible illumination. Bismuth sulfide (Bi_2_S_3_) is a semiconductor material with safe and non-toxic properties, high absorption coefficient, low band gap (~1.3 eV), abundant reserves and layered structure, which is widely available [21,22,23]. This material has emerged as an attractive material for supercapacitor, photodetectors and gas sensors [22,23,24]. Consequently, the integration of Bi_2_S_3_ with SAW devices under photoactivation conditions offers the potential for the development of a highly sensitive, rapid H_2_S sensor operating at room temperature.

In light of the aforementioned considerations, one-dimensional (1D) Bi_2_S_3_ nanoribbon structures were synthesized in this work through a straightforward solvothermal method. A light-activated SAW H_2_S sensor based on Bi_2_S_3_ nanoribbons was successfully designed and prepared. It is noteworthy that the Bi_2_S_3_ nanoribbons overlap with each other to form a reticulated film, which brings a large number of adsorption sites for the adsorption of H_2_S molecules. The removal of long-chain organic ligands from the Bi_2_S_3_ surface by ligand exchange facilitates the contact of H_2_S molecules with Bi_2_S_3_ nanoribbons, improving the sensor response. Furthermore, light activation has been shown to generate additional carriers of the Bi_2_S_3_ nanoribbons, which subsequently combine with oxygen present in the air to form adsorbed oxygen. This additional adsorbed oxygen is also involved in the gas-sensitive reaction, which improves the response of the sensor and reduces the response/recovery time. The sensor exhibited a response to 10 ppm H_2_S under yellow light activation that was approximately equal ~ two times greater than the response observed in a dark environment. The sensor response/recovery time is only 27 s/21 s for 10 ppm H_2_S under light activation. In addition, the sensor also provides excellent selectivity and reliable repeatability. The efficacy of the light-activated SAW H_2_S sensor based on Bi_2_S_3_ design in enhancing the response/recovery speed and sensitivity of gas sensors is well documented.

## 2. Materials and Methods

### 2.1. Chemicals and Instruments

Triphenyl bismuth (C_18_H_15_Bi, 98%), dibenzyl disulfide (C_6_H_5_CH_2_SSCH_2_C_6_H_5_, 98%) and oleyl amine (OLA) were purchased from Shanghai Aladdin Biochemical Technology Co., Ltd. (Shanghai, China). Polyvinyl pyrrolidone ((C_6_H_9_NO)_n_, PVP, MW ≈ 29,000) was purchased from Macklin Biochemical Technology Co., Ltd. (Shanghai, China). The hydrogen gas, nitrogen dioxide, hydrogen sulfide gases and ammonia gases with 2% concentration were provided by Gold Valley Gas Co., Ltd. (Shenzhen, China).

The microscopic surface morphology of Bi_2_S_3_ films were characterized using field emission scanning electron microscopy (FE-SEM, Supra 55 Sapphire, Zeiss, Baden-Württemberg, Germany). The crystal structures of the Bi_2_S_3_ nanoribbons were investigated using a high-resolution transmission electron microscope (HR-TEM, Tecnai G2 F20, Hillsboro, OR, USA) and an X-ray diffraction (XRD, X’pert pro, PANalytical, Almere, The Netherlands) with Cu Kα radiation source (40 kV and 40 mA). An X-ray photoelectron spectroscopy (XPS, Escalab 250Xi, Thermo Fisher, Dartford, UK) was used to investigate the chemical state of the Bi_2_S_3_-sensitive films.

### 2.2. Finite Element Analysis and Preparation of SAW Devices

A simplified three-dimensional finite element (FEA) model of the SAW delay line device was successfully achieved using COMSOL Multiphysics 6.0 software. The model is comprised three modules: Piezoelectric Effect module, Solid Mechanics module and Electrostatics modules [25,26]. A frequency-based solver is used to solve the total displacement [27]. As shown in Appendix A, the computational domain of the model consists of a Y-128° cut LiNbO_3_ piezoelectric material and a pair of interdigital transducers (IDT) electrodes and a perfect match layer (PML). The width of the SAW model (*x*-axis direction) is equal to the device wavelength (λ = 16 μm). The electrodes are designed with a width and spacing of 4 μm, and the electrodes are 80 nm in thickness. A 25 μm PML is applied to the bottom boundary of the model, while all boundaries are left free. Considering the computational volume of the model, the depth of the piezoelectric substrate and electrodes in the *y*-axis direction was set to 1/8 of the device wavelength (2 μm). A mapping approach is used to minimize computational complexity while ensuring simulation accuracy. As shown in Appendix A, the two-dimensional computing domain of S_21_ signal of SAW model consists of a Y-128° cut LiNbO_3_ (12,080 μm × 120 μm) coated with a 20 μm PML on the lower, left and right edges of the mode. All the boundary conditions are free boundary conditions. On the upper surface of the piezoelectric substrate, the left end is the input IDTs, and the right end is the output IDTs. The input-output IDTs consist of 20 pairs of fingers with an electrode width and spacing of 4 μm and a thickness of 80 nm. Input IDT plus 1 V sinusoidal signal, output IDT in floating potential condition.

SAW devices were fabricated using conventional photolithography and lift-off processes, based on our previous work [28]. The SAW device consists of 50 pairs of input and output fingers. The wavelength of the SAW device is 16 µm and the ratio of metal electrode width to slit width is 1:1.

### 2.3. Preparation of Bi_2_S_3_ Nanoribbons

A simple and straightforward solvothermal method was utilized to prepare one-dimensional Bi_2_S_3_ nanoribbons [29]. As demonstrated in Figure 1a, the triphenyl bismuth (C_18_H_15_Bi, 1.2 mM) and dibenzyl disulfide (C_6_H_5_CH_2_SSCH_2_C_6_H_5_, 1.2 mM) were dissolved in 8 mL of OLA to yield solution A. Subsequently, 0.8 g of polyvinylpyrrolidone (PVP) was ultrasonically dissolved in 52 mL of anhydrous ethanol to obtain solution B. Finally, the above A and B solutions were stirred and mixed, and then transferred to a Teflon-lined autoclave reactor and reacted for 8 h at 200 °C. Subsequent to the conclusion of the reaction, the product was repeatedly washed three times using anhydrous ethanol. The final product was uniformly dispersed in ethanol solvent to obtain a 50 mg/mL solution of Bi_2_S_3_ nanoribbons.

### 2.4. Fabrication and Measurement of SAW H_2_S Sensors

An amount of 50 μL of Bi_2_S_3_ nanoribbon solution was covered the central sensitive region (the delay region between input IDT and output IDT) of the SAW delay line device, ensuring complete coverage. Then, Bi_2_S_3_ was uniformly deposited on the surface of the SAW device by spin-coating at a rotational velocity of 1000 rpm for a duration of 90 s. The above steps were repeated to spin-coat two layers of Bi_2_S_3_ nanoribbon-sensitive films. Subsequently, a solution of Pb(NO_3_)_2_ in methanol (10 mg/mL) and pure methanol was deposited on the SAW sensor in an alternating manner, followed by a spin-coating at 1000 rpm for 100 s. This process was repeated thrice to ensure the removal of ligands that had been capping the surface of the Bi_2_S_3_ films to obtain a Pb(NO_3_)_2_-treated SAW H_2_S sensor. Similarly, Mg(NO_3_)_2_-treated, NaNO_2_-treated and Cu(NO_3_)_2_-treated SAW H_2_S sensors can be prepared separately.

The gas test system is illustrated in Figure 1b. The SAW sensor is interconnected with a network analyzer via coaxial RF cable. The computer collects the SAW sensor signals and outputs them in real time through a test program. The test data of SAW devices are collected by network analyzer, transmitted and recorded in real time through GPIB bus and computer test system. The frequency change before and after adsorption of H_2_S is directly output through the computer. The test chamber is equipped with LED light sources of varying wavelengths (395 nm, 465 nm, 520 nm, 590 nm and 625 nm), which are powered by a signal generator. During the light-activated gas-sensitive test, the optical power density was 20 μW/cm^2^. The gas sensitivity test is conducted in a static mode with a chamber volume of 18 L. For example, 9 mL of 2% H_2_S gas is injected into the chamber through a syringe, and a 10 ppm H_2_S test environment is obtained. The entire process is conducted within a fume hood, with temperature of 24 ± 1 °C and a relative humidity of 60%RH.

## 3. Results

### 3.1. FEA Simulations of SAW Devices

As illustrated in Figure 2b, the simulated S_21_ signal from the SAW device exhibits an optimal work frequency of 243.1 MHz, accompanied by an insertion loss of −17.52 dB. The displacement field distribution at the work frequency (Figure 2a) confirms the Rayleigh wave propagation pattern, which is characterized by elliptical particle motions that are primarily concentrated near the device surface. This indicates that the main displacement of the surface particles occurs in the *z*-axis and the device is sensitive to surface perturbations, which facilitates the achievement of highly sensitive detection of small gas molecules. The insertion loss profile of the SAW device is illustrated in Figure 2c. The center frequency was found to be 241.0 MHz, and the insertion loss was −16.3 dB. The minor discrepancies observed between the simulated and experimental outcomes can be primarily attributed to the inaccuracies in the thickness of the deposited electrodes during the preparation stage and the simplification of the finite element model. The insertion loss of the device was then compared before and after deposition of Bi_2_S_3_ (Figure 2c). After Bi_2_S_3_ deposition, the insertion loss changed from −16.3 dB to −24.7 dB (~0.84 dB), and the center frequency decreases from 241.0 MHz to 240.6 MHz, which is mainly due to the increase in the surface mass loading of the device.

### 3.2. Characterization of Bi_2_S_3_ Nanoribbons

Figure 3a shows the X-ray diffraction pattern of Bi_2_S_3_ nanoribbons, where the peaks with 2θ of 15.80°, 17.58°, 22.39°, 24.93°, 28.60°, 35.58°, 39.89°, 43. 12°, 47.33°, 48.27°, 62.68° and 66.90° correspond to the peaks at (020), (120), (130), (211), (240), (141), (250), (530), (470) and (470) of Bi_2_S_3_, respectively. Furthermore, the XRD patterns of the samples exhibited a strong resemblance to the characteristic peaks of the standard PDF#17-0320 [22], and there was an absence of any other impurity peaks in the results. This finding suggests that the prepared Bi_2_S_3_ nanoribbons are not only highly crystalline but also extremely pure. As demonstrated in Figure 3b, Bi_2_S_3_ nanoribbons exhibit pronounced absorption in the visible light spectrum, ranging from 300 to 700 nm. Therefore, these nanoribbons are anticipated to facilitate the development of highly sensitive and light-activated gas sensors within the visible light band.

To further investigate the chemical valence states and corresponding chemical compositions of the Bi_2_S_3_ nanoribbon, the samples were analyzed by XPS. The characteristic peaks of Bi 5d, S 2s, C 1s, Bi 4d, O 1s and Bi 4p can be clearly observed from the full XPS spectrum of the Bi_2_S_3_ sample in Figure 3c. In addition, the peaks of Bi 4f overlapped to some extent with the peak positions of S 2p, which was also confirmed by the high-resolution Bi 4f and S 2p XPS spectra (Figure 3d). In particular, the S 2p spectrum could be divided into two characteristic peaks of S 2p_1/2_ and S 2p_3/2_ with positions at 161.6 eV and 160.5 eV, respectively, which are attributed to the S element in Bi_2_S_3_ materials [23]. Meanwhile, the two characteristic peaks in the Bi 4f profile with positions at 163.0 eV and 157.8 eV are attributed to Bi 4f_5/2_ and Bi 4f_7/2_, respectively, suggesting that the +3 valent Bi in the Bi_2_S_3_ nanoribbons dominates the synthesized material.

SEM images of Bi_2_S_3_ nanoribbons are shown in Figure 4a–c. It can be clearly observed that the prepared Bi_2_S_3_ exhibits a typical 1D nanoribbon structure. The 1D Bi_2_S_3_ nanoribbons possess a large specific surface area, and the interlocking Bi_2_S_3_ nanoribbons form a reticulated film. These features enable a large number of adsorption sites for gas molecules, which is favorable for the adsorption of gas molecules to increase the gas-sensitive response. The tight binding of their overlapping is also conducive to the effective transfer of carriers between the materials, which is conducive to the rapid adsorption and desorption of gases. As illustrated in Figure 4h, the EDS mapping images of the Cu(NO_3_)_2_-treated Bi_2_S_3_ film demonstrates that Bi and S are distributed evenly, with a small amount of Cu resulting from the Cu(NO_3_)_2_ ligand substitution.

The HR-TEM images of Bi_2_S_3_ nanoribbons are displayed in Figure 4d,e. The low-magnification HR-TEM image (Figure 4d) confirms that the 1D nanostructure of Bi_2_S_3_ nanoribbons is primarily attributable to the highly anisotropic orthorhombic phase of Bi_2_S_3_, which facilitates preferential growth along the nanoribbon axis. The high magnification HR-TEM image (Figure 4e) demonstrates the lattice striations of Bi_2_S_3_. The inset illustrates the high-quality, ordered lattice striations of Bi_2_S_3_, with the lattice spacing of 0.199 nm corresponding to the (002) crystallographic plane of Bi_2_S_3_. Furthermore, the selected area electron diffraction (SAED) (Figure 4f) demonstrates the highly crystalline nature of the Bi_2_S_3_, displaying clear diffraction points for the (020), (301) and (002) crystal planes. These results are in agreement with the XRD test results, suggesting that high-quality 1D Bi_2_S_3_ nanoribbons have been successfully prepared.

### 3.3. Sensing Performance of SAW H_2_S Sensors

It is noteworthy that the surface of Bi_2_S_3_ nanoribbons synthesized via solvothermal method is covered with abundant long-chain surfactants (PVP and OLA), which play a crucial role in the formation of the nanoribbons and result in a well-dispersed solution with good processability [30]. Conversely, the organic long chains of OLA and PVP present on the surface of Bi_2_S_3_ nanoribbons have been shown to impede the adsorption of gas molecules, thereby hindering carrier transport within the material and resulting in suboptimal gas-sensitive properties [11]. The exchange of long-chain organic ligands with other short-chain ligands, such as inorganic salts, is a facile process. As demonstrated by the FT-IR measurements of Bi_2_S_3_ nanoribbon films treated with inorganic salts solution (see Figure 4g), the aliphatic C-H stretching vibrational peaks at 2856~2921 cm^−1^ and 1747 cm^−1^ in the untreated Bi_2_S_3_ nanoribbon films correspond to the organic long chains in OLA and PVP, respectively [29]. In contrast, these characteristic peaks were significantly attenuated after NaNO_2_, Mg(NO_3_)_2_, Pb(NO_3_)_2_ and Cu(NO_3_)_2_ ligand substitution treatments, which significantly indicated that most of the long-chain ligands were successfully removed by ligand exchange treatments with the above inorganic salts. Removal of insulating organic ligands covering the Bi_2_S_3_ surface enhances target gas adsorption and rapid charge transfer.

In order to compare the improvement of sensor performance by ligand substitution of different inorganic salt solutions, Bi_2_S_3_ nanoribbon H_2_S sensors treated with NaNO_2_, Zn(NO_3_)_2_, Pb(NO_3_)_2_ and Cu(NO_3_)_2_ solutions were prepared, respectively. The responses of all sensors were then subjected to testing at varying concentrations of H_2_S, ranging from 2 ppm to 50 ppm. The real-time response curves of the sensors without ligand substitution and after NaNO_2_, Mg(NO_3_)_2_ and Pb(NO_3_)_2_ ligand substitution for different concentrations of H_2_S are shown in Figure 5a–d.

The results show that the operating frequency of all sensors decreases after gas adsorption, which is mainly due to the change in mass loading caused by the gas adsorption process. In addition, all treated sensors showed good response recovery curves at both low and high H_2_S gas concentrations, which can be attributed to the excellent performance of the Bi_2_S_3_ nanoribbon material. The absence of ligand exchange of the Bi_2_S_3_ nanoribbon SAW sensor resulted in a frequency shift of −5.8 kHz for 50 ppm H_2_S gas and a response of only about −1 kHz for 2 ppm H_2_S. This is mainly due to the use of organic solvents during the synthesis process, resulting in the attachment of long-chain groups to the Bi_2_S_3_ nanoribbons, which hinders the adsorption of the target gases and leads to a weakened response. As a result, the responses of the ligand exchange-treated sensors were all higher than those of the untreated Bi_2_S_3_ sensor. This phenomenon can be attributed primarily to the ligand exchange which induced the sensitive films to be able to contact the H_2_S gas molecules more efficiently, which increased the adsorption of H_2_S to improve the response of the sensors. The responses of the SAW sensors were −6.26 kHz, −6.323 kHz, −11.298 kHz and −13.548 kHz for 50 ppm H_2_S gas, based on NaNO_2_, Zn(NO_3_)_2_, Pb(NO_3_)_2_ and Cu(NO_3_)_2_ treatments, respectively. The detailed comparison of the responses at other concentrations is shown in Appendix A.

The real-time curve of the sensor response to different concentrations of H_2_S gas after Cu(NO_3_)_2_ ligand exchange treatment is shown in Figure 6a. The response of the sensor with different ligand exchange treatments compared with the untreated Bi_2_S_3_ nanoribbon sensor for different H_2_S gas concentrations is shown in Figure 6b. For different concentrations of H_2_S, the sensor exhibited an approximately linear response under 10 ppm. And with the increase of H_2_S concentration, the response of the sensors gradually tends to saturation, which may be due to the limited number of gas adsorption sites in the sensitive films leading to H_2_S adsorption saturation. Among them, as demonstrated in Figure 6b, the Cu(NO_3_)_2_-treated sensor exhibits the maximum response. This phenomenon may be attributed to the Cu ions exhibiting a preference for the adsorption of H_2_S gas, which has been confirmed in previous reports [31]. Therefore, the subsequent study was conducted on the basis of the Cu(NO_3_)_2_ solution treatment.

The response of the Cu(NO_3_)_2_-treated sensor to 10 ppm H_2_S gas was tested under 20 μW/cm^2^ light (395 nm, 465 nm, 520 nm, 590 nm and 625 nm) as well as in a dark environment, respectively. Figure 7a presents the real-time response curves of the Cu(NO_3_)_2_-treated SAW sensor to 10 ppm H_2_S gas at different wavelength light activation. Figure 7b shows the corresponding line plots of response comparison at different wavelength light gains. The results show that the response of the sensor to H_2_S gas varies with light wavelength, and the light-activated responses are all enhanced compared to the dark environment. Specifically, the response value of the sensor in the dark under 10 ppm H_2_S gas was −6.1 kHz, while the frequency shift of the sensor operating at 590 nm light-activated was improved to −11.5 kHz. This represents a substantial enhancement of 188.5% in performance compared to the dark condition.

The response/recovery time of a gas sensor is also an indispensable indicator for evaluating its performance. The response time of a gas sensor is defined as the time required for the response to reach 90% of the maximum value when the sensor is exposed to the target gas environment; the recovery time is defined as the time required for the response to recover from the maximum value to 90% of the initial state [32]. The response/recovery times of the Bi_2_S_3_ nanoribbons SAW sensor for 10 ppm H_2_S gas at 590 nm light-activated are 27 s/21 s, as shown in Figure 7c. In a similar manner, the response/recovery times of the SAW sensor with different wavelengths light-activated for 10 ppm H_2_S gas were analyzed (see Figure 7d). It was found that the response/recovery speeds of the sensor with different wavelengths light-activated were improved compared with those in the dark environment. In particular, the response/recovery becomes faster and faster as the wavelength of the activated light increases. This enhancement can be attributed chiefly to the variation in the number of photogenerated carriers excited by different wavelengths of light at 20 μW/cm^2^. The increase in response speed is due to light activation changing the adsorption kinetics on the surface of the sensitive material. This process leads to a decrease in the gas adsorption reaction potential and an acceleration of target gas adsorption and desorption.

The response of the SAW gas sensor with 590 nm light-activated to H_2_S gas (0.5–50 ppm) is shown in Figure 8a. When exposed to H_2_S gas, the center frequency of the sensor decreases to a negative frequency shift. When the sensor is exposed to air again, the center frequency of the sensor gradually returns to the initial frequency value. It is clear that the frequency shifts of the sensor gradually increase as the target H_2_S concentration increases. The sensor’s response to 50 ppm H_2_S can reach a maximum of −18.2 kHz. The sensor displays a response of −2.8 kHz even when the H_2_S concentration is reduced to 0.5 ppm. This demonstrates the excellent performance of the sensor, which can reach sub-ppb levels.

A comparison of the responses of the sensor with light-activated and dark shows that the response is enhanced by 134% (50 ppm H_2_S), as shown in Figure 8b. Furthermore, a 191% improvement in response was observed at a concentration of 10 ppm, with a 163% improvement even at a low concentration of 2 ppm. The enhancement of the light-activated response is particularly pronounced for low concentrations of H_2_S below 10 ppm, which facilitates the detection of low concentrations of H_2_S.

The relationship between H_2_S concentration (0.5~50 ppm) and frequency shifts of the sensor is shown in Figure 8c. The frequency response of the sensor in the low concentration range of H_2_S from 0.5 to 8 ppm shows good linearity with R^2^ of 0.981. The good linearity of the sensor facilitates the commercial application and development of the sensor. As the H_2_S concentration increased (greater than 10 ppm), the frequency change in the sensor leveled off, which was limited by the number of adsorption sites of sensitive area leading to saturation of H_2_S adsorption. The theoretical detection limit (LOD) of the sensor can be calculated based on the signal-to-noise ratio with the following formula [33]:(1)$$LOD\left(ppm\right)=3\frac{SD}{S}$$
where the standard deviation (SD) of the 300 data points of the baseline is 0.0344 and S is the slope of the linear fit line of 1.08306 in Figure 8c. According to Equation (1), the LOD is calculated to be about 95.2 ppb. A comparative evaluation of H_2_S sensing capabilities between the SAW gas sensor developed in this study and previously reported SAW-based devices is presented in Table 1. The results demonstrate that the Bi_2_S_3_ nanoribbon-functionalized sensor exhibits superior performance characteristics, particularly with respect to enhanced frequency response and accelerated response/recovery time.

The repeatability of the sensor was evaluated by repeated exposure to 10 ppm H_2_S, as shown in Figure 8d. The four frequency shifts of the sensor are almost constant with less than 2% variation in the response value, suggesting a good repeatability of the sensor. Figure 8e reveals the sensor selectivity by comparing the response of various gases commonly, including H_2_S, NO_2_, NH_3_, H_2_ methanol, ethanol and acetone gases. The response of the above interfering gases is less than 10% of the frequency response of the 10 ppm H_2_S gas, indicating that the SAW sensor based on Bi_2_S_3_ nanoribbons has a good selectivity for H_2_S gas.

Subsequently, the response of the Bi_2_S_3_-based SAW sensor to 10 ppm H_2_S was tested under varying relative humidity conditions (30~80%). As illustrated in Appendix A, the frequency response (Δf) of the SAW sensor increases with the rise in RH values (30~80%). This phenomenon can be attributed to the significant adsorption of H_2_S and H_2_O molecules on the sensitive area under high relative humidity, which leads to an increase in the mass loading of the SAW sensor and amplifies the frequency shifts [35,38].

### 3.4. Sensing Mechanism of SAW H_2_S Sensors

The gas-sensing mechanism can be elucidated in terms of the chemical reaction that occurs between the sensitive material and H_2_S. In air, oxygen molecules are adsorbed on the surface of the sensitive material and extract electrons from the conduction band of Bi_2_S_3_ films, formatting of adsorbed oxygen ions $O_{2(ads)}^{-}$, $O_{\left(ads\right)}^{-}$ and $O_{\left(ads\right)}^{2-}$ on the surface of the Bi_2_S_3_ nanoribbon [40]. The specific reaction process is shown in Figure 9a and Equations (2)–(5). Electron abstraction from the surface of the Bi_2_S_3_ nanoribbons leads to the formation of an electron depletion layer in the n-type Bi_2_S_3_ nanoribbons.(2)$$O_{2(gas)}\to O_{2(ads)}$$(3)$$O_{2(ads)}+e^{-}\to O_{2(ads)}^{-}$$(4)$$O_{2(ads)}^{-}+e^{-}\to {2O}_{\left(ads\right)}^{-}$$(5)$$O_{\left(ads\right)}^{-}+e^{-}\to O_{\left(ads\right)}^{2-}$$(6)$${2H}_{2}S+{3O}_{2(ads)}^{-}\to {2SO}_{2}+{2H}_{2}O+{3e}^{-}$$(7)$$H_{2}S+{3O}_{\left(ads\right)}^{-}\to {SO}_{2}+H_{2}O+{3e}^{-}$$(8)$$H_{2}S+{3O}_{\left(ads\right)}^{2-}\to {SO}_{2}+H_{2}O+{6e}^{-}$$

Upon exposure to H_2_S gas, the H_2_S molecules react with the adsorbed oxygen on the surface of the Bi_2_S_3_ nanoribbons, resulting in the release of electrons back to the Bi_2_S_3_ surface (Figure 9b). The process can be explained in detail by Equations (6)–(8), and the schematic is shown in Figure 9b. This process leads to changes in the mass and conductance of the sensitive film on the surface of the SAW sensor.

In the case of SAW gas sensors, the gas molecules react with the sensitive material on the surface of the sensor, thereby affecting the propagation of the SAW. The frequency shift of the SAW sensor is attributed to a combination of mass loading effect, acoustic and electrical effects, and viscoelastic effect, as follows [9,41]:(9)$$\frac{\Delta f}{f_{0}}≅\frac{\Delta v}{v_{0}}=-C_{m}f_{0}\Delta \left(\rho_{s}\right)+4C_{e}f_{0}\Delta \left(hG^{\prime }\right)-\frac{K^{2}}{2}\Delta \left(\frac{1}{1+{\left(\frac{v_{0}c_{s}}{\sigma_{s}}\right)}^{2}}\right)$$
where Δf signifies the frequency shift, $f_{0}\;$ denotes the center frequency, Δv and $v_{0}$ represent the change in wave velocity and the initial wave velocity, $C_{m}$ and $C_{e}$ are the coefficients of mass sensitivity and elasticity of the material. The change in mass per unit area is denoted by $\rho_{s}$, the thickness of the film is represented by h, and the shear modulus is indicated by $G^{\prime }$. The electromechanical coupling coefficient is denoted by $K^{2}$, the capacitance per unit length of the substrate is denoted by $c_{s}$, and the conductivity of the film is denoted by $\sigma_{s}$.

According to Equation (9), an increase in the conductivity and an increase in the mass of the sensitive film leads to a decrease in the center frequency of the SAW (negative frequency shift). In order to distinguish between the respective effects of mass loading effect and acoustic electrical effects on frequency shift, a 50 nm Au film was deposited on the sensitive area of the SAW delay line device. The highly conductive metal films eliminate the frequency shift of the SAW device due to acoustoelectric effects. Additionally, the mass loading effect remains unaffected by the Au film. As illustrated in Appendix A, the responses of the sensors with and without Au film to 10 ppm H_2_S were −7.81 kHz and −11.57 kHz, respectively. This finding indicates that the response exhibited by Bi_2_S_3_-based SAW sensors to H_2_S is the result of a combination of mass loading and acoustic electric effects, with the mass loading effect contributing 67.5% of the total response.

## 4. Conclusions

In summary, we have prepared a fast and highly sensitive light-activated SAW H_2_S sensor based on 1D Bi_2_S_3_ nanoribbons. The optimal performance of the Cu(NO_3_)_2_-treated sensor was determined by investigating ligand exchange with different inorganic salts. The Cu(NO_3_)_2_-treated sensor exhibited a response value of −6.08 kHz for 10 ppm H_2_S, which is 203% of the untreated sensor. Furthermore, the performance of the SAW sensor was evaluated under light activation with different wavelengths. When exposed to 10 ppm H_2_S under yellow light activation (590 nm), the sensor demonstrated a frequency shift of −11.57 kHz, which corresponded to 191% of the response in the dark. Furthermore, when the H_2_S concentration was reduced to 500 ppb, the sensor exhibited a response of −2.8 kHz with a theoretical detection limit of 31.8 ppb. Therefore, light-activated SAW H_2_S sensors based Bi_2_S_3_ nanoribbons present a promising avenue for enhancing the overall performance of gas sensors and the application of nanomaterials in the domain of gas sensing technology.

## Figures and Tables

**Figure 1 sensors-25-01122-f001:**
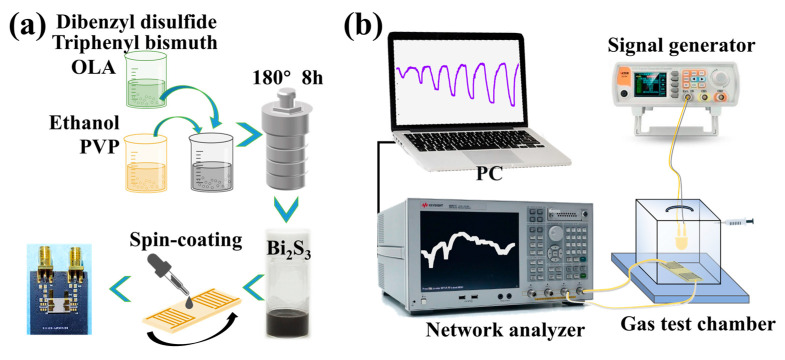
(**a**) Schematic of SAW H_2_S sensor preparation; (**b**) diagram of light-activated gas-sensitive test system.

**Figure 2 sensors-25-01122-f002:**
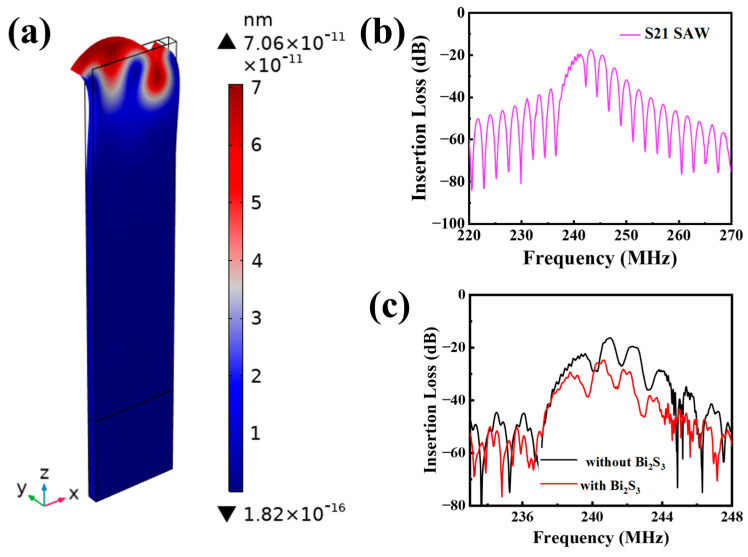
(**a**) Displacement of the SAW device under peak frequency excitation; (**b**) FEA-simulated S_21_ signals of the SAW device; (**c**) S_21_ signals of the fabricated SAW device before and after coated Bi_2_S_3_ films.

**Figure 3 sensors-25-01122-f003:**
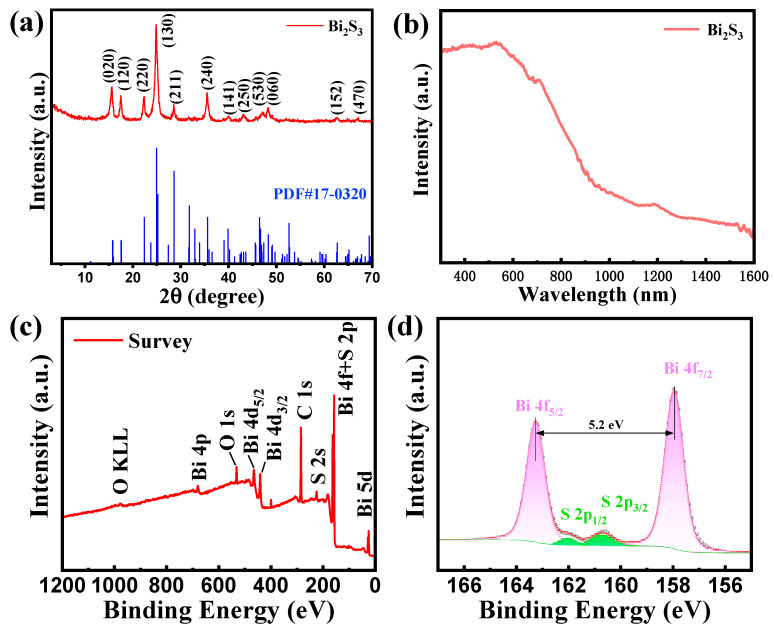
(**a**) XRD patterns of Bi_2_S_3_ nanoribbons; (**b**) absorption spectra of Bi_2_S_3_ nanoribbons; (**c**) overall XPS spectra of Bi_2_S_3_ nanoribbons; (**d**) focused XPS spectra of Bi 4f and S 2p.

**Figure 4 sensors-25-01122-f004:**
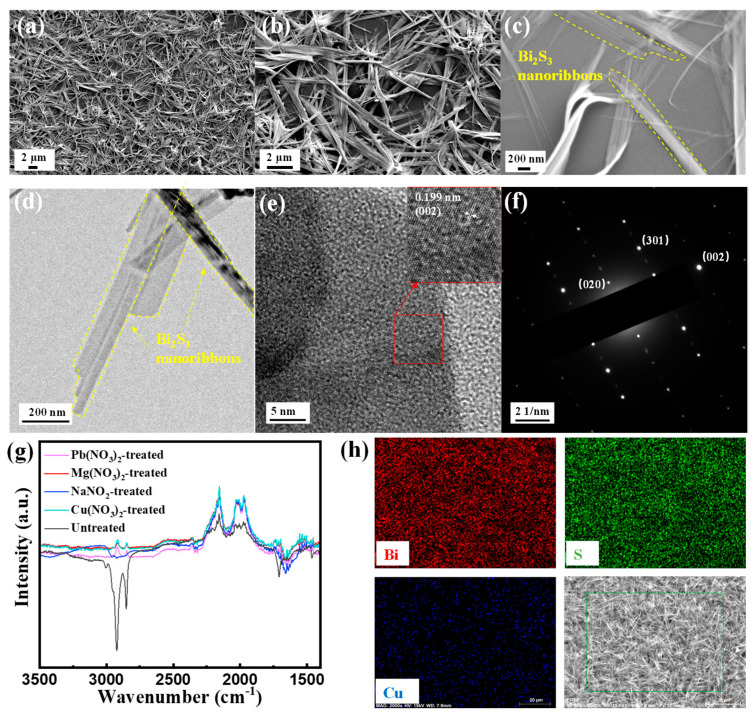
(**a**–**c**) SEM images of Bi_2_S_3_ at different magnifications; (**d**,**e**) HR-TEM images of Bi_2_S_3_; (**f**) corresponding SAED patterns of Bi_2_S_3_; (**g**) FTIR spectra of the Bi_2_S_3_ films treated with different ligands; (**h**) the EDS map scanning analysis of Bi, S and Cu elements.

**Figure 5 sensors-25-01122-f005:**
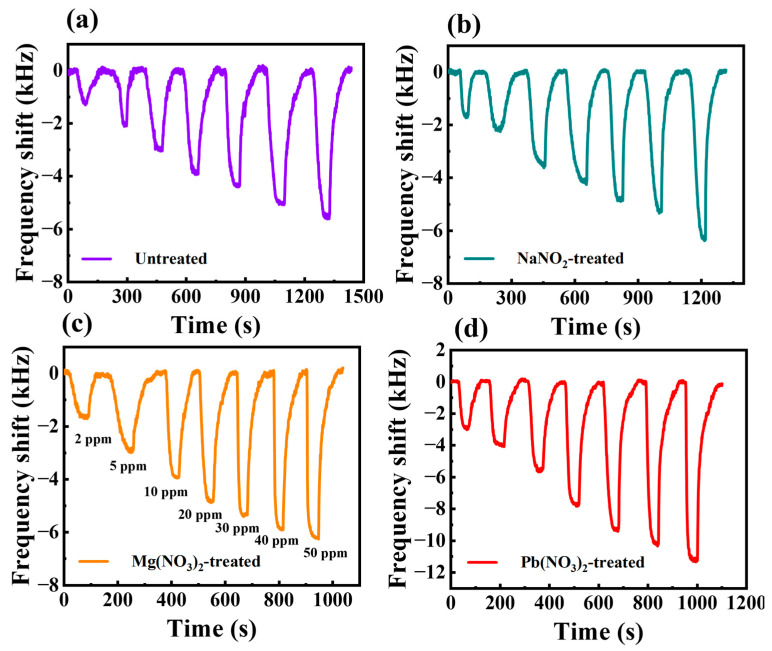
Real-time response curves of the sensor to different concentrations of H_2_S for (**a**) unliganded substitution, (**b**) NaNO_2_ solution treatment, (**c**) Mg(NO_3_)_2_ solution treatment, (**d**) Pb(NO_3_)_2_ solution treatment.

**Figure 6 sensors-25-01122-f006:**
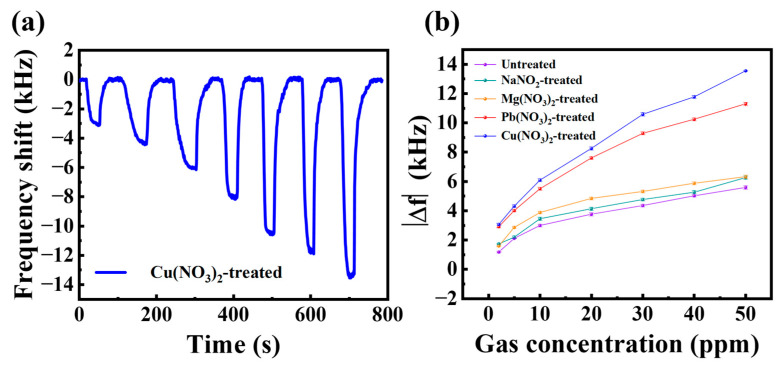
(**a**) Real-time response of the sensor treated with Cu(NO_3_)_2_ to different concentrations of H_2_S gas; (**b**) comparison of the response of different ligand substitutions to different concentrations of H_2_S gas.

**Figure 7 sensors-25-01122-f007:**
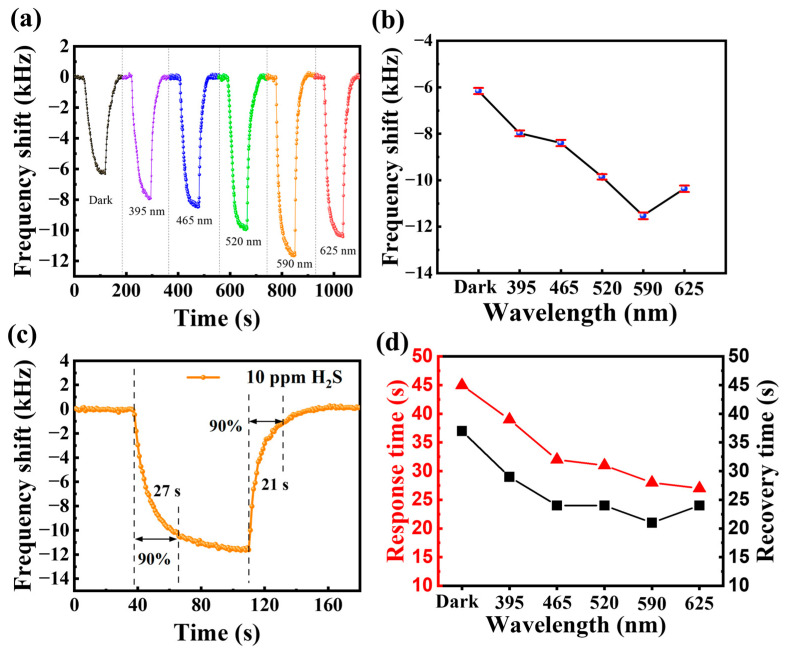
(**a**) Real-time response curve of Bi_2_S_3_ nanoribbon SAW sensor to 10 ppm H_2_S gas at different wavelength light-activated; (**b**) comparison plot of the responses of Bi_2_S_3_ nanoribbon SAW sensor to 10 ppm H_2_S gas at different wavelength light-activated; (**c**) response/recovery times to 10 ppm H_2_S at 590 nm light-activated; (**d**) comparison of sensor response/recovery times to 10 ppm H_2_S at different wavelength light-activated.

**Figure 8 sensors-25-01122-f008:**
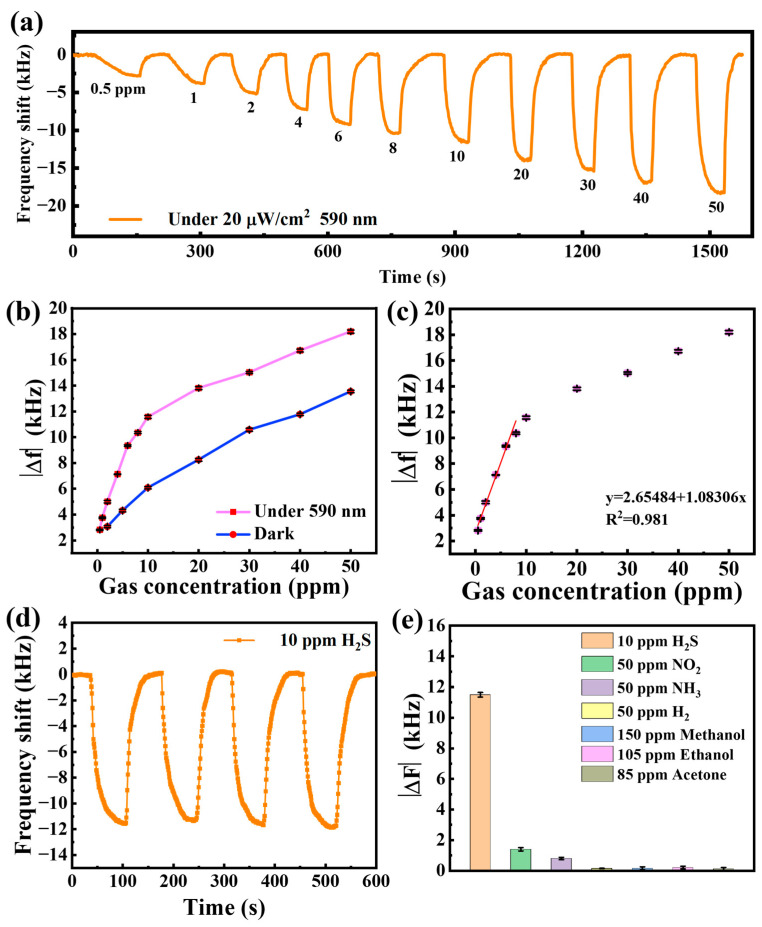
(**a**) Real-time response curve of the sensor to different concentrations of H_2_S at 590 nm light-activated; (**b**) comparison of sensor response to H_2_S under 590 nm light-activated and darkness; (**c**) frequency response of H_2_S at different concentrations under 590 nm light-activated; (**d**) repeatability of the sensor for 10 ppm H_2_S gas; (**e**) selectivity of the sensor.

**Figure 9 sensors-25-01122-f009:**
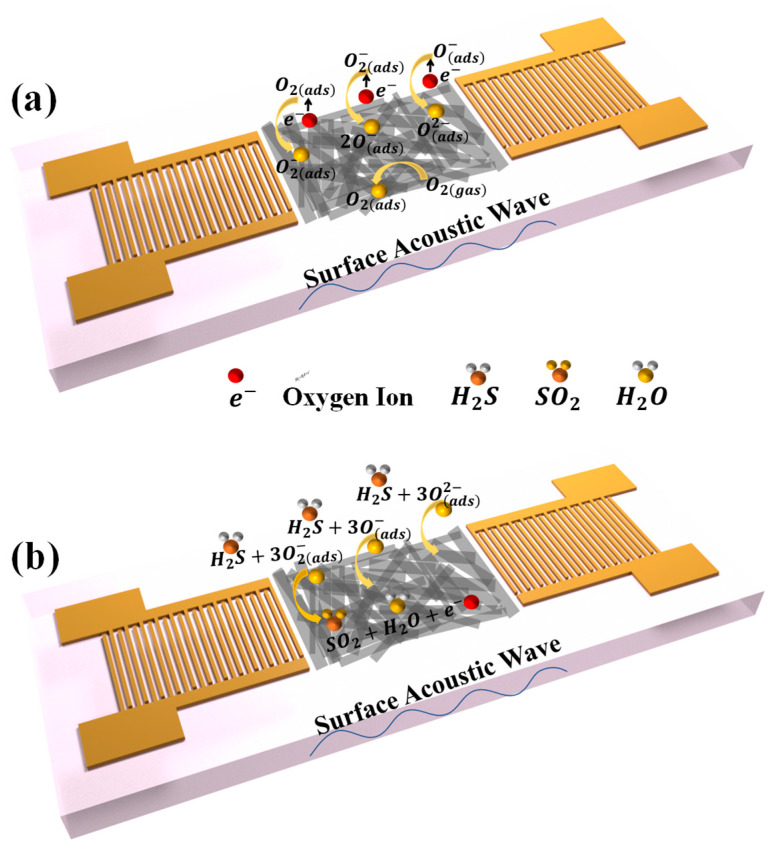
Schematic diagram of gas-sensitive mechanism of the sensor (**a**) in air (**b**) in H_2_S.

**Table 1 sensors-25-01122-t001:** Comparison of sensitivity and response time of SAW H_2_S gas sensor.

Sensitive Materials	Operating Frequency (MHz)	Concentration (ppm)	|Δf| (kHz)	Response Time (s)	Ref.
Fe_2_O_3_/SiO_2_	200	1	0.8	<200	[34]
CuO-Al_2_O_3_	200	1	15	>1000	[35]
CuO	200	2	9	>3000	[36]
CuO/TiO_2_	200	10	34	531	[37]
CuO@V_2_C	163	20	39	54	[38]
CuO	200	4	30	>200	[39]
Bi_2_S_3_	241	10	11.5	27	This work

## Data Availability

The original contributions presented in this study are included in the article/Appendix A. Further inquiries can be directed to the corresponding author(s).

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
