# Peer review of "Light-Activated Room Temperature Surface Acoustic Wave H_2_S Sensor Based on Bi_2_S_3_ Nanoribbons"

_sensors, 2025, doi:10.3390/s25041122_

Round 1
Reviewer 1 Report
Comments and Suggestions for Authors
The experiments design and discussion in this paper can be supported the conclusion, but the following points need to be added:
1. The SO2 gas should be tested for the clarifying of selectivity.
2. The effect of humidity should be discussed.
3.Please compare to other reports of H2S SAW sensor.
4.Please clarify the innovation of the sensing materials synthesized process.
Reviewer 2 Report
Comments and Suggestions for Authors
The paper presents the results of a study on a new configuration of SAW sensor/sensitive layer for H2S detection. The sensitive layer is made of Bi2S3 nanoribbons, which can be treated with various chemical compounds to slightly enhance its performance (sensitivity, speed, etc.).
The study is interesting but falls into a long line of similar papers published by many research groups over the past 25 years. Typically, these studies involve testing a new sensitive layer and comparing the results to a simple FEM model (usually COMSOL). It is typically demonstrated that the frequency response varies when the gas concentration changes, within the ppm range up to a few tens of ppm. The issues of selectivity (the sensor may also react to other gases) and comparison with optical sensors (which generally perform much better and are significantly faster) are usually overlooked.
This paper does not seem to deviate from this trend. Although significant effort has undoubtedly been put into fabricating the sensitive layer and performing the measurements, the sensitivity and reaction speed results are very similar to those obtained with hundreds of other layers (see the literature over the past 25 years). The FEM model itself is a standard and well-known tool, which is not even used to predict, even approximately, the sensor's sensitivity. At this point, showing that this classic model, available with COMSOL, can calculate the transfer function of a simple Delay Line quite accurately doesn't add much value.
To increase the interest of this paper, it should compare (perhaps in a table?) the performance results with those obtained by many other authors (see review papers?). Additionally, a comparison of the SAW/Bi2S3 sensor's performance with optical H2S sensors would be useful. Finally, this paper cannot be considered complete without at least a simple study of selectivity. If this sensor is also sensitive to other gases, it loses much of its relevance (the development of specific sensitive layers aims primarily to reduce the sensitivity to other gases to zero, although coupling the sensor with some kind of filtering system is always an option. However, this would complicate the system and, in any case, make the solution unattractive for practical applications).
The citations should be better chosen. The ones used give the impression (though I may be mistaken) of a rushed bibliographic search that also completely overlooks papers older than a few years. This can give the impression of a young and rapidly growing research field, while this is absolutely not the case. Once again, this type of paper has been regularly published by many groups for over 25 years.
Specific comments to the authors:
Line 110: Typo - "As shown in Figure. S1."
The COMSOL model used for this study is old and very well-known, having been applied in numerous studies. Please provide at least 1-2 citations of previous work based on this model, to avoid giving the false impression that you developed it. Moreover, the elementary cell model presented in Figure 2a is insufficient for computing the full transfer function. Please present the entire model and explain, with more detail, how you computed the transfer function.
The resolution of Figures 1, 3, 4 and 9 must be (strongly) improved. In particular, Figure 4 needs improvement, as it shows important and useful results.
Line 196: "It can be clearly observed that the prepared Bi2S3 exhibits a typical 1D nanoribbon structure." Please improve. Add some additional information on the figures to help the reader identify the nanoribbons (e.g., arrows, boxes, etc.)
Reviewer 3 Report
Comments and Suggestions for Authors
The article is devoted to the development of a sensor for determining the concentration of hydrogen sulfide. The sensor's operation is based on the dependence of the SAW velocity on the concentration of H2S due to its absorption by a thin film of Bi2S3, which is pretreated with Cu(NO3) and illuminated with yellow light. As a result, a very good sensitivity of 31.8 ppb was obtained, which is an undoubted achievement of the authors of the work. However, when describing the work of the SAW device itself, the authors made a number of inaccuracies. For example, in line 114 it is indicated that the thickness of the electrodes of IDT is 80 microns, which is 20 times greater than their width, which cannot be. This needs to be clarified and corrected in order for the article to be accepted for publication. In addition, it is described that the Bi2S3 film, which is located between IDT changes the center frequency. But the film is outside of the IDT and cannot change their central frequency. In this case, the SAW velocity changes under the action of the film, which leads to a change in the phase of SAW, and hence to a change in the conditions of interference of the triple-pass signal with the main signal, which often leads to a shift in the maxima and minima of the S21parametr. The direct frequency shift of the maxima (minima) is also not shown under the influence of changes in the concentration of hydrogen sulfide. All this needs to be fixed for the article to be published.

Round 2
Reviewer 3 Report
Comments and Suggestions for Authors
The article can now be published